# Beyond the Shot: Rethinking Cinematography Understanding with Foundational Skill Evaluation

## Abstract

Cinematography understanding refers to the ability to recognize not only the visual content of a scene but also the cinematic techniques that shape narrative meaning. This capability is attracting increasing attention, as it enhances multimodal understanding in real-world applications and underpins coherent content creation in film and media. As the most comprehensive benchmark for this task, ShotBench spans a wide range of cinematic concepts and VQA-style evaluations, with ShotVL achieving state-of-the-art results on it. However, our analysis reveals that ambiguous option design in ShotBench and ShotVL's shortcomings in reasoning consistency and instruction adherence undermine evaluation reliability, limiting fair comparison and hindering future progress. To overcome these issues, we systematically refine ShotBench through consistent option restructuring, conduct the first critical analysis of ShotVL's reasoning behavior, and introduce an extended evaluation protocol that jointly assesses task accuracy and core model competencies. These efforts lead to ShotBench++, a refined and expanded benchmark that enables more reliable assessment and fosters future advances in cinematography understanding. The benchmark and code will be publicly released.

## 1 Introduction

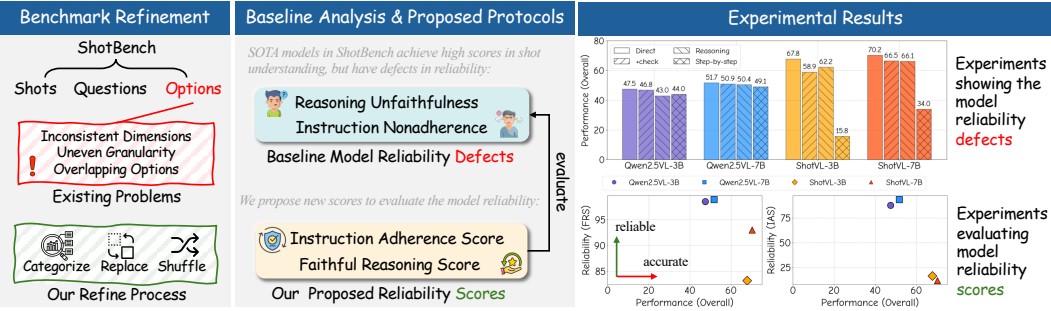

Figure 1: Overview of our work. We first analyze and refine the options in ShotBench to address their inconsistencies, then examine state-of-the-art models and reveal their reliability defects. Based on these findings, we propose a new evaluation protocol and demonstrate its effectiveness through comprehensive experiments.

Cinematography understanding represents a specialized form of multimodal reasoning that requires models to analyze not only the visual content of a scene but also the cinematic techniques that shape narrative construction. This capability goes beyond conventional video recognition by demanding fine-grained perception of camera movements, lighting conditions, shot composition, framing strategies, and other stylistic choices that filmmakers employ to guide audience attention and evoke emotion. Mastering such understanding is crucial for capturing the creative intent behind visual storytelling, rather than merely describing surface-level content. As multimodal large language

models advance toward real-world applications in creative industries, education, and media analysis, the ability to reliably evaluate cinematography understanding becomes increasingly critical. Robust benchmarks in this domain are essential not only for measuring task performance but also for probing deeper reasoning skills, ensuring that progress in model development aligns with the complexities of narrative-driven visual communication.

ShotBench (Liu et al., 2025b) has emerged as the primary benchmark for this task, offering over 3,500 expert-annotated multiple-choice questions across eight cinematographic dimensions. The benchmark has enabled systematic evaluation of model capabilities and established performance baselines, with ShotVL achieving state-of-the-art results across multiple categories. However, the reliability of these evaluations depends fundamentally on the quality of the underlying benchmark design and the robustness of the evaluated models.

Our systematic analysis reveals two categories of issues that may compromise current evaluation practices. First, examination of ShotBench's multiple-choice design shows inconsistencies in option granularity and evaluation dimensions. Questions intended to assess lighting conditions, for example, sometimes mix directional descriptors with intensity descriptors, creating scenarios where multiple answers could be defensible. These ambiguities can obscure genuine model capabilities and introduce confounding factors into performance comparisons.

Second, detailed investigation of ShotVL's behavior reveals discrepancies between reported performance and underlying reasoning reliability. Through controlled experiments measuring consistency between reasoning traces and final answers, we observe that model predictions are not always grounded in the stated reasoning process. Additionally, ShotVL exhibits significant performance degradation when required to follow structured instruction formats, suggesting limitations in instruction adherence that standard accuracy metrics do not capture.

These findings indicate that current evaluation may provide an incomplete picture of model capabilities in cinematography understanding. High accuracy scores may mask fundamental issues with reasoning consistency and instruction following, potentially affecting the validity of model comparisons and limiting insights for future improvements. To address these limitations, we introduce ShotBench++, which refines the original benchmark through systematic reorganization of multiple-choice options and incorporates expanded evaluation protocols.

Our contributions can be summarized as below:

- **Benchmark Refinement.** We redesign the multiple-choice option sets in ShotBench by enforcing consistent granularity, unified evaluation dimensions, and mutual exclusivity. This renders a coherent and reliable dataset for evaluating cinematography understanding.

- **Critical Analysis of State-of-the-Art Baselines.** We conduct the first in-depth study of ShotVL, the reported state-of-the-art on ShotBench, and reveal fundamental weaknesses in reasoning, prompt adherence, and output consistency, challenging the validity of its benchmark superiority.

- **Expanded Evaluation Protocol.** We augment ShotBench with a new protocol that jointly assesses task-specific performance and core model competencies, providing a more balanced and robust framework for fair comparison and future progress in this emerging field. Together, these contributions establish **ShotBench++**, a refined and extended benchmark for cinematography understanding.

## 2 RELATED WORK

### 2.1 CINEMATOGRAPHY UNDERSTANDING

Early works on automatic film analysis have studied sub-tasks such as shot type classification, scene segmentation, and cut recognition, with MovieShots (Rao et al., 2020) and MovieNet (Huang et al., 2020) providing basic taxonomies but focusing mainly on shot size and camera movement. Later benchmarks like CameraBench (Lin et al., 2025) and CineTechBench (Wang et al., 2025) expanded the scope by incorporating camera angle, motion primitives, and richer evaluation dimensions. However, these efforts still fall short of capturing the full spectrum of cinematic language, and even ShotBench—the first attempt at a comprehensive framework—faces challenges in option design

and baseline reliability. Our work addresses these gaps by refining ShotBench's construction and evaluation protocol toward a more principled framework.

## 2.2 MULTIMODAL UNDERSTANDING

Significant advancements in large language models (LLMs) (Touvron et al., 2023; Brown et al., 2020; Chowdhery et al., 2023) have inspired the development of multimodal large language models (MLLMs) (Li et al., 2024b; Yin et al., 2023; Bai et al., 2024). Early MLLM efforts, such as LLaVA (Liu et al., 2024a), MiniGPT-4 (Zhu et al., 2023), and InstructBLIP (Dai et al., 2023), demonstrate notable multimodal understanding capabilities. To integrate LLMs into multimodal domains, these studies explored projecting features from a pre-trained modal-specific encoder, such as CLIP (Radford et al., 2021), into the input space of LLMs, enabling multimodal understanding and reasoning within the transformer backbone. There are various design choices of MLLM (McKinzie et al., 2024; Tong et al., 2024; Wu et al., 2025; Liu et al., 2025a) in vision encoders, feature alignment adapters, and datasets.

## 2.3 BENCHMARKING MLLMS

Vision-Language Models (VLMs) (Bai et al., 2025; Team et al., 2023; Zhu et al., 2025; Liu et al., 2024c; 2025c; Li et al., 2024a; Zhang et al., 2024b) have shown strong progress across perception, reasoning, and multi-modal tasks, with benchmarks ranging from general-purpose (e.g., MMBench (Liu et al., 2024b), MMVU (Zhao et al., 2025)) to domain-specific evaluations such as logical reasoning, spatial reasoning, egocentric video, scientific figures, and visual programming (Xiao et al., 2024; Ramakrishnan et al., 2024; Mangalam et al., 2023; Yang et al., 2024; Roberts et al., 2024; Wang et al., 2024; Hu et al., 2025; Zhang et al., 2024a). Yet, none explicitly target cinematography understanding, an essential dimension of visual storytelling. ShotBench was proposed to address this gap but suffers from limitations in design and baseline robustness. Building on it, we propose refined dataset construction, critical baseline analysis, and an expanded evaluation protocol for a stronger foundation in benchmarking MLLMs for cinematic language.

## 3 BENCHMARK REFINEMENT

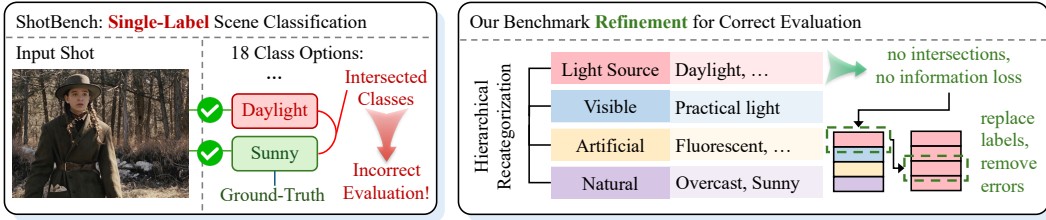

Figure 2: Refining dataset options by introducing a finer-grained taxonomy and replacing inconsistent choices in ShotBench. This ensures that options within each question are mutually exclusive and of consistent granularity.

In this section, we discuss the improperly designed options in ShotBench and explain how we modify the data to improve the dataset's fairness and accuracy, all without altering the original annotations.

During our careful review of ShotBench, we identified inconsistencies in the design of multiple-choice options. While all candidate answers nominally belong to the same category, they are described from heterogeneous dimensions, which undermines the mutual exclusivity of the options. This design flaw introduces ambiguity into the evaluation process, making the results less reliable and potentially unfair. For instance, in the lighting condition task, terms such as side light, backlight, top light, and underlight characterize the direction of illumination, whereas high contrast and low contrast describe its contrast level, and hard light and soft light capture its intensity. Since these attributes come from fundamentally different dimensions, they inevitably overlap. Mixing such heterogeneous descriptors within the same option set not only confuses models but also weakens the

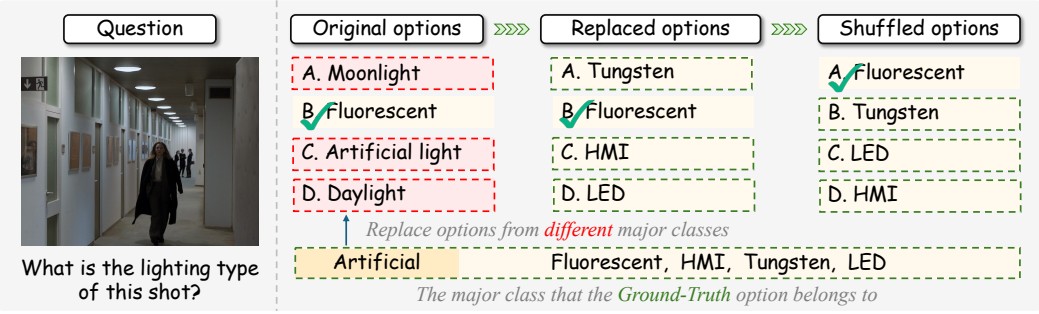

Figure 3: Refinement Case. This figure shows how inconsistent lighting type labels are improved for a benchmark dataset. We first map the ground-truth option to its corresponding refined category, remove options from mismatched categories, replace them with alternatives from the same category, and finally randomize the order to ensure fairness.

validity of the evaluation. Formally, we denote a question $q$ with an option set

$$O_q = \{o_{q,1}, o_{q,2}, \ldots, o_{q,k}\}, \qquad a_q \in O_q,$$

where $a_q$ is the annotated answer. Each option $o \in O_q$ belongs to a subclass $S_d$ determined by its descriptive dimension $d$. However, in the original benchmark, it is possible that

$$\exists\, o_i, o_j \in O_q : M(o_i) \neq M(o_j),$$

meaning that options come from different dimensions and thus violate the principle of mutual exclusivity.

To resolve this issue, we refined and standardized the option design. Specifically, we required that every option in the refined set $O'_q$ is drawn from the same subclass as the annotated answer, i.e.,

$$\forall\, o_i, o_j \in O'_q, \quad M(o_i) = M(o_j).$$

Based on this taxonomy, we systematically revised each question by first locating the annotated answer's subclass $S_{M(a_q)}$ and then constructing the refined option set as

$$O'_q = \{a_q\} \cup \text{Sample}(S_{M(a_q)} \setminus \{a_q\}, k-1).$$

Afterwards, we applied random shuffling to eliminate ordering bias. In cases where the subclass contained only one element (i.e., $|S_{M(a_q)}| = 1$), the multiple-choice format was replaced with a binary classification task,

$$y_q \in \{0, 1\},$$

which preserves evaluation validity without introducing artificial distractors.

In total, we revised 961 questions and constructed an improved version of ShotBench. Based on this benchmark, we re-evaluated two representative models, namely the state-of-the-art baseline ShotVL and the pre-tuned Qwen model. The results, summarized in Tab. 1, highlight the effectiveness of our modifications and demonstrate that the refined benchmark enables a more reliable assessment of model performance under consistent and mutually exclusive option settings.

The lighting condition task illustrates both the problems in the original dataset and the benefits of our refinement. Previously, its framework combined physically defined categories such as Firelight with functionally defined ones such as Practical light, which led to severe misclassifications. For example, the original benchmark showed a 16.7% confusion rate between Artificial light and Practical light. By contrast, our refined benchmark adopts a standardized scheme in which all options are consistently defined by physical properties. This leads to substantial improvements, with accuracy on broad categories like Overcast rising to 97.3%. At the same time, the re-evaluation also reveals models' current limitations, including almost no accuracy for LED (0.0%) and persistent confusion between physically similar sources such as HMI and Sunny. These results demonstrate that our refinement not only ensures a fairer and more systematic evaluation, but also transforms the benchmark into a powerful diagnostic tool that exposes fine-grained weaknesses of existing models and highlights the challenge of aligning nuanced visual cues with precise technical terminology.

Table 1: Model performance on the refined ShotBench across eight tasks: lens source LS, lighting type LT, lighting condition LC, shot framing SF, shot size SS, camera angle CA, shot composition SC, and camera movement CM. Overall denotes the average across tasks.

| Model | LS | LT | LC | SF | SS | CA | SC | CM | Overall |
|---|---|---|---|---|---|---|---|---|---|
| Qwen2.5VL-3B | 35.8 | 52.6 | 57.7 | 78.7 | 49.7 | 40.7 | 40.1 | 29.7 | 47.5 |
| Qwen2.5VL-7B | 44.6 | 55.6 | 48.9 | 69.7 | 63.3 | 48.6 | 45.7 | 37.7 | 51.7 |
| ShotVL-3B | 60.5 | 64.0 | 67.4 | 91.0 | 79.4 | 68.1 | 60.8 | 51.3 | 67.8 |
| ShotVL-7B | **61.8** | **66.2** | **65.7** | **91.5** | **81.7** | **72.8** | **62.2** | **59.7** | **70.2** |

## 4 BASELINE MODEL ANALYSIS

In this section, we present a systematic investigation of state-of-the-art baseline models on Shot-Bench. Our goal is to uncover the root causes of their unexpected behaviors and to design controlled experiments that validate these findings. By identifying and formalizing such failure modes, we integrate them into ShotBench to provide a more comprehensive and reliable benchmark.

### 4.1 REASONING FAITHFULNESS

ShotVL-3B was generally able to produce explicit reasoning traces when prompted. However, closer inspection revealed frequent inconsistencies between the reasoning process and the final answer. Our statistical analysis uncovered two characteristic failure modes: (1) cases where the reasoning was logically sound and even reached the correct solution, yet the final answer was wrong; and (2) cases where the reasoning process was erroneous or incoherent, yet the model nevertheless produced the correct answer. These discrepancies indicate that the model's predictions are not consistently grounded in its reasoning, thereby undermining the faithfulness and trustworthiness of its outputs. Such inconsistencies represent a critical limitation, as they obscure whether correct answers are derived from genuine reasoning or from coincidental correlations.

**Qualitative Analysis.** As illustrated in Fig. 4, we instruct the model to produce its reasoning process under the ¡think¿ tag and the final prediction under the ¡answer¿ tag. This setup requires the model to first articulate its reasoning and then provide an answer consistent with that reasoning. However, when given such instructions, ShotVL frequently generates responses where the ¡think¿ and ¡answer¿ outputs contradict each other. In some cases, the reasoning is correct while the final answer is wrong, and in other cases, the reasoning is flawed but the final answer happens to be correct. These inconsistencies raise concerns about the model's reasoning faithfulness and cast doubt on whether its outputs genuinely reflect reliable reasoning capabilities.

**Quantitative Analysis.** To further validate these observations, we designed targeted experiments to systematically examine the alignment between reasoning and final answers. As shown in Tab. 2, we introduce the +check evaluation, where Qwen3-2B is employed as an automated verifier to compare the reasoning trace within the `<think></think>` tags against the final answer within the `<answer></answer>` tags. If the two are inconsistent—meaning the answer cannot be logically derived from the reasoning steps—the output is deemed incorrect. Under this evaluation, ShotVL-3B exhibits a substantial drop of 8.9 points in overall accuracy, from 68.3% to 59.0%, whereas Qwen2.5VL-3B and Qwen2.5VL-7B show negligible changes. This marked decline indicates that many of ShotVL-3B's correct answers do not faithfully follow from its reasoning, confirming that the model often produces superficially correct outputs that are not grounded in its own reasoning process.

### 4.2 REASONING INSTRUCTION ADHERENCE

To better understand the limitations of current multimodal reasoning systems, we conducted a detailed analysis of ShotVL-3B and ShotVL-7B. Our examination shows that ShotVL-7B consistently struggles to follow explicit step-by-step reasoning prompts, even when such instructions are clearly specified in the system prompt. Instead of producing intermediate reasoning steps as required, the

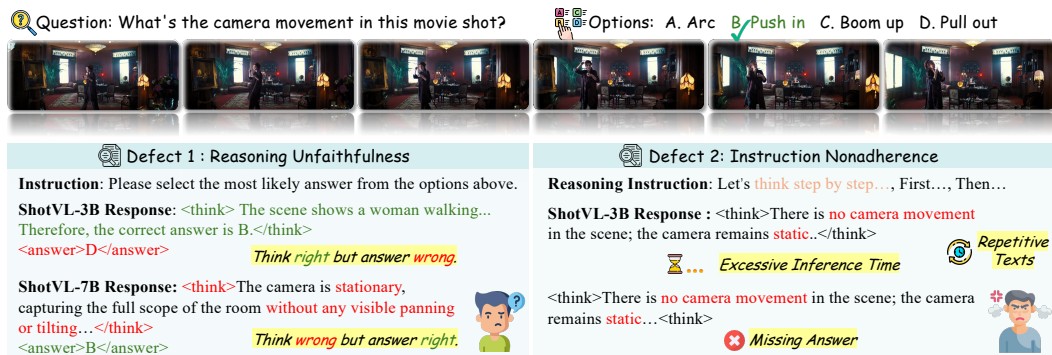

Figure 4: Model Analysis. This figure shows two main defects of ShotVL models: reasoning un-faithfulness, with frequent mismatches between reasoning and answers, and poor instruction adherence, where prompts are ignored in favor of long repetitive outputs.

Table 2: Experimental results of models after consistency check. We evaluate all model outputs for consistency between reasoning and final answers, treating mismatched cases as incorrect. The Qwen series shows almost no performance drop, while ShotVL suffers a notable decrease, indicating weaker reasoning faithfulness.

| Model | LS | LT | LC | SF | SS | CA | SC | CM | Overall |
|---|---|---|---|---|---|---|---|---|---|
| Qwen2.5VL-3B | 35.8 | 52.6 | 57.7 | 78.7 | 49.7 | 40.7 | 40.1 | 29.7 | 47.5 |
| +check | 35.6 | 48.4 | 56.7 | 78.7 | 49.1 | 40.2 | 40.1 | 29.7 | 46.8 ↓ 0.7 |
| Qwen2.5VL-7B | 44.6 | 55.6 | 48.9 | 69.7 | 63.3 | 48.6 | 45.7 | 37.7 | 51.7 |
| +check | 44.2 | 55.3 | 48.3 | 69.2 | 62.9 | 47.3 | 45.1 | 35.3 | 50.9 ↓ 0.8 |
| ShotVL-3B | 60.5 | 64.0 | 67.4 | 91.0 | 79.4 | 68.1 | 60.8 | 51.3 | 67.8 |
| +check | 52.8 | 56.6 | 59.1 | 82.0 | 70.3 | 60.4 | 50.9 | 39.9 | 58.9 ↓ 8.9 |
| ShotVL-7B | 61.8 | 66.2 | 65.7 | 91.5 | 81.7 | 72.8 | 62.2 | 59.7 | 70.2 |
| +check | 57.3 | 62.7 | 63.1 | 91.5 | 79.4 | 67.5 | 57.4 | 53.2 | 66.5 ↓ 4.7 |

model often bypasses them and directly outputs final answers. This pattern persists across multiple prompt reformulations, suggesting a disconnect between the model's nominal scale and its practical ability to execute natural language reasoning protocols. Such limitations reduce interpretability and raise concerns about the model's reliability in scenarios that demand strict adherence to structured instructions.

**Qualitative Analysis** Figure 5 illustrates this issue with a case study on camera movement recognition. The correct label is "Static shot," and both ShotVL-7B and Qwen2.5VL-7B were prompted to follow a four-step chain-of-thought reasoning process with a strict output format. While ShotVL-7B correctly identified the static shot in free-form reasoning, its prediction was marked incorrect because it failed to conform to the required structure and formatting. In contrast, Qwen2.5VL-7B not only arrived at the correct answer but also adhered closely to the reasoning steps and formatting rules, explicitly enumerating the options, analyzing the visual evidence, and structuring the output as instructed. This comparison highlights that the evaluation of advanced multimodal models should extend beyond answer accuracy to include compliance with complex instructions and transparent reasoning—capabilities that are increasingly essential for ensuring reliability and controllability.

**Quantitative Analysis** The quantitative results in Tab. 3 further reinforce these observations. ShotVL-7B achieves strong overall accuracy of 70.2 under direct prompting, but performance declines to 66.1 with reasoning prompts and drops sharply to 34.0 under step-by-step prompts. The degradation is particularly evident in tasks requiring structured reasoning, such as lighting type (66.2 to 19.5), camera angle (72.8 to 21.1), and camera movement (59.7 to 15.1). At the same time, in-

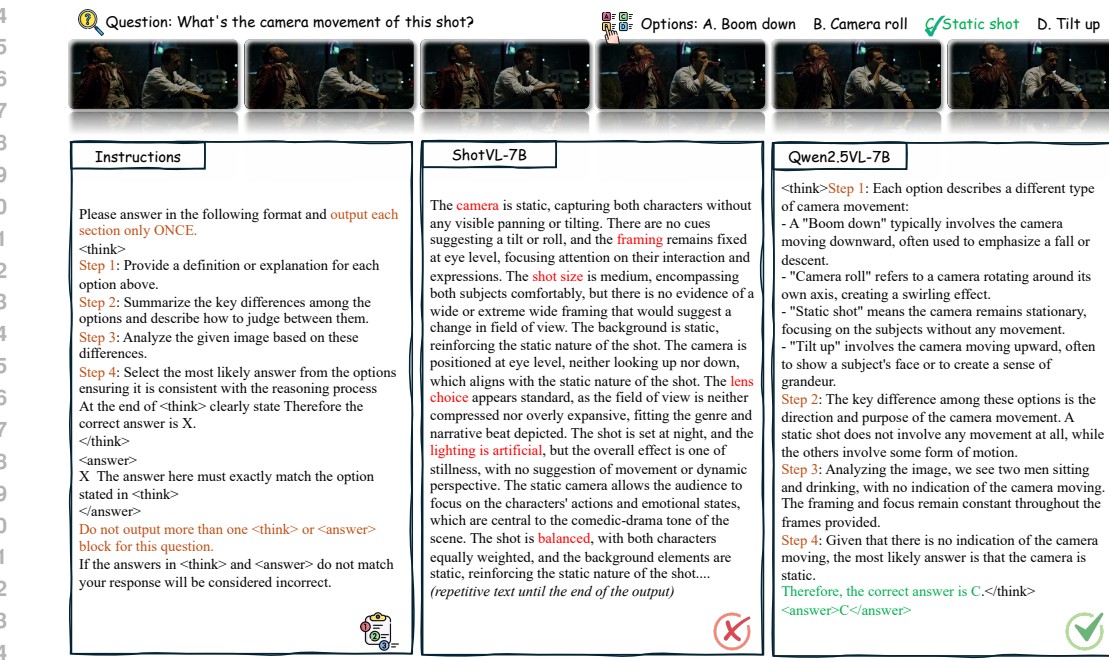

Figure 5: Instruction adherence case. This case shows the instruction adherence of different models. When given a demonstration-based prompt, ShotVL fails to follow the instructions and produces disorganized reasoning, whereas Qwen accurately follows the format, outputting each step and the final answer as required.

ference time increases substantially, from 0:04 under direct prompting to 0:37 with step-by-step prompts, indicating inefficiency when handling reasoning-specific instructions.

In contrast, Qwen2.5VL models exhibit a more stable pattern and in some cases even improve under reasoning-oriented prompts. For example, Qwen2.5VL-7B achieves 51.7 overall accuracy under direct prompting and remains comparable at 50.4 with reasoning prompts, with subsets such as Long-term Temporal increasing from 55.6 to 57.0 and Camera Motion from 37.7 to 38.2. Under step-by-step prompting, overall accuracy is 49.1, while Scene Factors improves from 69.7 to 75.7. These results suggest that Qwen can effectively leverage explicit reasoning instructions to refine its predictions in specific contexts, demonstrating stronger robustness and instruction-following capability.

Taken together, these findings reveal a fundamental distinction between the two models. ShotVL achieves high performance under straightforward direct prompts but suffers substantial degradation once reasoning-specific instructions are introduced, exposing a clear limitation in its language and reasoning competence. By contrast, Qwen, although less accurate in absolute terms, shows resilience and in some cases benefits from structured prompting. This contrast underscores that ShotVL's apparent strength under simple settings does not translate into robust reasoning ability, highlighting notable deficiencies in its foundational capabilities and raising concerns about its suitability for reasoning-intensive multimodal tasks.

# 5 EVALUATION PROTOCOL

In this section, we build on the issues identified in the previous analysis and introduce a new evaluation protocol tailored to address these shortcomings. We integrate this protocol into ShotBench to form a more comprehensive benchmark. Using the refined version of ShotBench, we then re-evaluated state-of-the-art models and conducted a detailed analysis of their experimental results.

Table 3: Performance of different models on the refined benchmark under reasoning and step-by-step prompts. Qwen remains stable across prompts, while ShotVL shows clear performance drops and higher time cost. Time cost is reported in hours:minutes (hh:mm) format.

| Model | LS | LT | LC | SF | SS | CA | SC | CM | Overall | Time cost |
|-------|------|------|------|------|------|------|------|------|---------|-----------|
| **Qwen2.5VL-3B** | | | | | | | | | | |
| Direct | 35.8 | 52.6 | 57.7 | 78.7 | 49.7 | 40.7 | 40.1 | 29.7 | 47.5 | 2:28 |
| Reasoning | 34.0 | 51.1 | 54.6 | 56.9 | 42.5 | 42.5 | 36.1 | 30.6 | 43.0 ↓4.5 | 9:25 |
| Step-by-step | 39.1 | 50.1 | 48.3 | 67.4 | 39.4 | 38.9 | 36.3 | 36.2 | 44.0 ↓3.5 | 9:26 |
| **Qwen2.5VL-7B** | | | | | | | | | | |
| Direct | 44.6 | 55.6 | 48.9 | 69.7 | 63.3 | 48.6 | 45.7 | 37.7 | 51.7 | 6:20 |
| Reasoning | 40.1 | 57.0 | 47.7 | 67.9 | 59.0 | 50.1 | 44.7 | 38.2 | 50.4 ↓1.3 | 12:15 |
| Step-by-step | 42.3 | 55.8 | 46.0 | 75.7 | 54.7 | 44.4 | 40.1 | 35.6 | 49.1 ↓2.6 | 17:52 |
| **ShotVL-3B** | | | | | | | | | | |
| Direct | 60.5 | 64.0 | 67.4 | 91.0 | 79.4 | 68.1 | 60.8 | 51.3 | 67.8 | 5:39 |
| Reasoning | 57.3 | 51.1 | 57.2 | 86.7 | 72.2 | 62.6 | 56.8 | 51.7 | 62.2 ↓5.6 | 18:31 |
| Step-by-step | 4.5 | 27.7 | 20.0 | 29.2 | 12.0 | 10.8 | 4.0 | 22.4 | 15.8 ↓52.0 | 42:49 |
| **ShotVL-7B** | | | | | | | | | | |
| Direct | 61.8 | 66.2 | 65.7 | 91.5 | 81.7 | 72.8 | 62.2 | 59.7 | 70.2 | 4:26 |
| Reasoning | 58.9 | 62.0 | 49.7 | 87.9 | 80.4 | 64.6 | 63.1 | 58.2 | 66.1 ↓4.1 | 13:46 |
| Step-by-step | 29.9 | 19.5 | 26.3 | 62.7 | 58.4 | 21.1 | 35.3 | 15.1 | 34.0 ↓36.2 | 37:47 |

## 5.1 METRICS

To systematize the empirical findings above, we formalize the exposed failure modes into three diagnostic evaluation protocols and integrate them as modular extensions of ShotBench. Each protocol targets a distinct dimension of model reliability and produces interpretable metrics that go beyond conventional accuracy reporting.

**Faithful Reasoning Score (FRS)** We define the Faithful Reasoning Score as the average consistency between the reasoning trace and the final answer. For each example, we assign a score of 1 if the conclusion in <think> matches the output in <answer>, and 0 otherwise. The overall metric is then computed as

$$\text{FRS} = \frac{1}{N} \sum_{i=1}^{N} g_i,$$

where $N$ is the number of evaluation samples. A higher FRS indicates that the model's final answers are more faithfully aligned with its own reasoning traces.

**Instruction Adherence Score (IAS)** We introduce the *Instruction Adherence Score (IAS)* to jointly evaluate instruction-following and answer correctness. For each input under a step-by-step prompt as shown in Fig. 5, we first use Qwen-3B as an automatic judge to verify whether the model output strictly follows the prescribed reasoning and formatting instructions. If not, the response is marked incorrect. Only outputs that both adhere to the instruction and provide the correct answer are considered fully correct. Formally, IAS is defined as the ratio between accuracy under this adhered-evaluation and the model's original accuracy:

$$\text{IAS} = \frac{\text{Acc}_{\text{adhered}}}{\text{Acc}_{\text{orig}}},$$

where $\text{Acc}_{\text{adhered}}$ denotes the accuracy requiring both instruction adherence and correct answers, and $\text{Acc}_{\text{orig}}$ denotes the original accuracy. A higher IAS reflects stronger and more reliable instruction-following ability.

Table 4: Performance and reliability of different models on the refined benchmark. Using our proposed evaluation, we find that Qwen achieves near-perfect reliability, significantly outperforming ShotVL, whose results are notably weaker, particularly in instruction adherence.

| Model | Performance | | | | | | | | | Reliability | |
|---|---|---|---|---|---|---|---|---|---|---|---|
| | LS | LT | LC | SF | SS | CA | SC | CM | Overall | FRS | IAS |
| Qwen2.5VL-3B | 35.8 | 52.6 | 57.7 | 78.7 | 49.7 | 40.7 | 40.1 | 29.7 | 47.5 | 98.5 | 87.8 |
| Qwen2.5VL-7B | 44.6 | 55.6 | 48.9 | 69.7 | 63.3 | 48.6 | 45.7 | 37.7 | 51.7 | **98.9** | **93.5** |
| ShotVL-3B | 60.5 | 64.0 | 67.4 | 91.0 | 79.4 | 68.1 | 60.8 | 51.3 | 67.8 | 83.2 | 16.4 |
| ShotVL-7B | **61.8** | **66.2** | **65.7** | **91.5** | **81.7** | **72.8** | **62.2** | **59.7** | **70.2** | 93.0 | 11.7 |

## 5.2 RESULTS

Using the protocol introduced in the previous section, we re-evaluated the models on the refined benchmark to systematically assess both performance and reliability. The results in Tab. 4 provide critical insights into state-of-the-art multimodal models. ShotVL-7B achieves the highest overall performance with a score of 70.2, yet its instruction adherence is only 19.7, revealing a substantial gap between raw accuracy and the ability to follow structured reasoning. Similarly, ShotVL-3B attains 67.8 overall accuracy, but exhibits low reliability and instruction adherence, indicating potential weaknesses in consistent reasoning and instruction-following. These observations suggest that high benchmark scores alone may mask fundamental deficiencies, which can affect downstream evaluation, comparison, and model improvement.

In contrast, Qwen2.5VL models show balanced and reliable behavior, achieving moderate overall accuracy while maintaining very high reliability. Qwen2.5VL-7B attains an instruction adherence score of 94.8 and failure rate stability of 98.9, whereas Qwen2.5VL-3B reaches 89.1 and 98.5. This demonstrates that Qwen consistently follows structured prompts and effectively leverages reasoning instructions, producing outputs that are both accurate and robust. In certain sub-tasks, such as long-term temporal reasoning and camera motion recognition, Qwen even improves under reasoning-specific prompts, highlighting its stability as a baseline model and providing a reliable foundation for future task-specific improvements without compromising core capabilities.

Taken together, these findings reveal a clear distinction between the two model families. ShotVL achieves high accuracy under simple prompts but suffers from low reliability and poor instruction adherence, exposing hidden weaknesses in reasoning and language capabilities. Qwen, by contrast, delivers stable, interpretable outputs and strong compliance with structured instructions. Crucially, these insights are enabled by our evaluation protocol, which jointly measures performance and reliability and exposes subtle failure modes overlooked by traditional benchmarks. Applying this protocol to cinematography understanding provides new perspectives for evaluating reasoning-intensive multimodal tasks, establishing rigorous standards for assessment, and guiding targeted improvements in advanced models.

## 6 CONCLUSION

In this work, we identify critical limitations in ShotBench and its state-of-the-art ShotVL baselines, including ambiguous multiple-choice options and fundamental weaknesses in reasoning, prompt adherence, and output consistency. To address these issues, we introduce **ShotBench++**, a refined and extended benchmark that enforces consistent option granularity, mutual exclusivity, and unified evaluation dimensions, while also incorporating a complementary evaluation protocol assessing both task-specific performance and core model competencies. Our in-depth analysis of ShotVL reveals overfitting to dataset artifacts and highlights the necessity of evaluating fundamental reasoning alongside benchmark scores. By providing a more reliable and comprehensive framework, ShotBench++ offers novel insights into cinematography understanding and sets a new standard for future research in developing models that truly capture cinematic form and intent.

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

Table 5: Performance of different models with perturb. This table verified the ShotVL-7B's defects in contextual robustness by adding irrelevant options in the question.

| Model | LS | LT | LC | SF | SS | CA | SC | CM | Overall |
|---|---|---|---|---|---|---|---|---|---|
| Qwen2.5VL-3B (Bai et al., 2025) | 35.8 | 52.6 | 57.7 | 78.7 | 49.7 | 40.7 | 40.1 | 29.7 | 47.5 |
| +1 perturb | 34.6 | 50.6 | 52.3 | 73.9 | 51.1 | 39.6 | 37.6 | 30.6 | 45.8 (-1.7) |
| Qwen2.5VL-7B (Bai et al., 2025) | 44.6 | 55.6 | 48.9 | 69.7 | 63.3 | 48.6 | 45.7 | 37.7 | 51.7 |
| +1 perturb | 44.6 | 55.8 | 46.7 | 70.6 | 62.1 | 48.6 | 48.0 | 37.5 | 51.7 (-) |
| ShotVL-3B (Liu et al., 2025b) | 60.5 | 64.0 | 67.4 | 91.0 | 79.4 | 68.1 | 60.8 | 51.3 | 67.8 |
| +1 perturb | 59.1 | 64.0 | 65.3 | 91.9 | 81.4 | 67.3 | 59.7 | 50.4 | 67.4 (-0.4) |
| ShotVL-7B (Liu et al., 2025b) | 61.8 | 66.2 | 65.7 | 91.5 | 81.7 | 72.8 | 62.2 | 59.7 | 70.2 |
| +1 perturb | 61.2 | 65.9 | 63.7 | 92.1 | 81.9 | 71.4 | 62.2 | 59.1 | 69.8 (-0.4) |

## A  APPENDIX

### A.1  LLM USAGE

I have used large language models just to polish my paper writing.

### A.2  OTHER EXPERIMENTS

We have also conducted other experiments on the reliability of models. The results are shown in Tab. 5. The results reveal several notable patterns. First, larger models generally achieve higher baseline performance, with ShotVL-7B attaining the highest overall accuracy. Second, the impact of adding a single irrelevant option varies across models. Qwen2.5VL-3B experiences a noticeable drop of 1.7 points, while Qwen2.5VL-7B remains largely unaffected, indicating that model size improves contextual robustness for this architecture. In contrast, ShotVL models, despite their strong overall performance, show small but consistent decreases under perturbation, suggesting that even state-of-the-art models are susceptible to subtle contextual changes. These findings highlight the importance of robustness evaluation when assessing model reliability in complex question-answering scenarios.

### A.3  PROMPT

In our experiments, we use a reasoning-style prompt designed to guide the model through a structured thought process. As shown in the example below, the prompt first presents the question and candidate options, then instructs the model to select the most likely answer while explicitly encouraging step-by-step reasoning. Specifically, the model is asked to output its thinking process in sequential steps before providing the final answer, which allows us to evaluate not only the correctness of the response but also the faithfulness of the model's reasoning.

**Prompt**

```
reasoning_prompt = (
    f"Question: {q}\n{opts_block}\n"
    "Please select the most likely answer from the options above."
    "Let's think step by step."
    "You should output the thinking process in step 1, step 2 and so on."
)
```

