# OpenReview forum: "Beyond the Shot: Rethinking Cinematography Understanding with Foundational Skill Evaluation"
_ICLR.cc/2026/Conference — ICLR 2026 Conference Withdrawn Submission_

### Official Review · Reviewer_sCg5 · 2025-10-30

**Soundness:** 3
**Presentation:** 3
**Contribution:** 2
**Rating:** 4
**Confidence:** 2

**Summary:**

This paper addresses the task of cinematography understanding—the ability of multimodal large language models to recognize not only the visual content in video shots but also the underlying cinematic techniques such as lighting, camera movement, composition, and framing. The authors identify two key issues in the existing benchmark ShotBench and its baseline model ShotVL: (1) Ambiguous and inconsistent multiple-choice design; (2) Baseline unreliability, including reasoning unfaithfulness and poor instruction adherence. To address these problems, the paper proposes: (1) ShotBench++, a refined version of ShotBench with consistent and mutually exclusive options; (2) New evaluation protocols introducing Faithful Reasoning Score (FRS) and Instruction Adherence Score (IAS) to jointly measure task accuracy and reliability; Comprehensive re-evaluation of ShotVL and Qwen2.5-VL models, showing that high task accuracy does not necessarily imply reliable reasoning or prompt adherence.

**Strengths:**

1. The authors identify concrete flaws in ShotBench and provide a clear taxonomy-based correction procedure, ensuring consistent option granularity and mutual exclusivity.

2. The discovery of reasoning unfaithfulness and instruction non-adherence is an important observation for the community. The use of structured prompts with <think> / <answer> tags is clear and reproducible.

3. These metrics provide valuable quantitative measures beyond accuracy, offering a new lens for evaluating reasoning consistency in MLLMs.

**Weaknesses:**

1. The primary contribution of this paper lies in refining the benchmark and introducing an evaluation protocol, rather than proposing a new modeling technique. The paper mainly focuses on diagnosing and reorganizing an existing dataset, rather than presenting a new dataset, algorithm, or representation framework.

2. Experiments are limited to two model families (ShotVL and Qwen2.5-VL). There is no evaluation on other MLLMs such as LLaVA-OneVision, InternVL3, or Gemini. This raises concerns about generality.

3. The FRS and IAS rely on automated verification using Qwen as a reasoning checker, which may introduce bias. There is no discussion of cross-validation using human judges or other LLMs to confirm consistency.

4. While the authors claim improved granularity and exclusivity, they do not measure inter-annotator agreement or human validation on the refined dataset. Thus, it remains unclear whether the new benchmark truly improves human interpretability or only aligns internally defined taxonomies.

**Questions:**

Please see Weakness

---

### Official Review · Reviewer_a5Uq · 2025-10-31

**Soundness:** 3
**Presentation:** 3
**Contribution:** 3
**Rating:** 4
**Confidence:** 4

**Summary:**

This paper presents ShotBench++, a refined and extended benchmark for cinematography understanding that goes beyond visual recognition to include cinematic techniques such as lighting, framing, and camera movement. The authors identify two major issues with the existing ShotBench and its associated model ShotVL: (1) ambiguous and inconsistent multiple-choice options, and (2) unreliable reasoning and instruction adherence in current state-of-the-art models.
To address these, they systematically restructure the benchmark options for consistency and introduce new diagnostic protocols, including Faithful Reasoning Score (FRS) and Instruction Adherence Score (IAS), to jointly assess performance and reliability. Experimental analyses on ShotVL and Qwen variants validate the proposed refinements and highlight meaningful behavioral differences between models.

**Strengths:**

- **Thorough analysis**: The paper systematically diagnoses flaws in both benchmark construction and model behavior, offering a clear motivation for ShotBench++.
- **Clear writing and structure**: The paper reads well and articulates its motivations and contributions in a logically coherent manner.
- And I think the proposed FRS and IAS metrics are clearly formulated and conceptually grounded in reliability and interpretability.

**Weaknesses:**

- **Dependence on existing datasets**: Most experiments are re-evaluations on ShotBench; there is limited evidence of generalization to new cinematography datasets or unseen domains.
- **Evaluation scope**: The study primarily compares two model families (ShotVL and Qwen). Broader evaluation across diverse MLLMs would make the findings more generalizable.

**Questions:**

- How sensitive are the new metrics (FRS, IAS) to the choice of the verifier model (e.g., Qwen-3B)? Would results change if another LLM were used as the judge?
- In constructing ShotBench++, were human experts involved to verify the correctness and coherence of the revised options?
- Could the proposed evaluation framework be used in a training-time diagnostic context, e.g., to improve reasoning consistency during fine-tuning?
- In the conclusion, you said: ''Our in-depth analysis of ShotVL reveals overfitting to dataset **artifacts** and highlights the necessity of evaluating fundamental reasoning alongside benchmark scores.'' And where did you provide evidence of **artifacts**?

PS: I am leaning toward raising my score if the authors can solve the weekness and the question.

---

### Official Review · Reviewer_58d7 · 2025-11-01

**Soundness:** 2
**Presentation:** 3
**Contribution:** 2
**Rating:** 2
**Confidence:** 4

**Summary:**

The paper targets at the field of Cinematography and VQA to evaluate how well can VLMs understand the fine-grained Cinematography techniques. The benchmark paper refined an existing dataset, and evaluated on Qwen2.5 and ShotVL two sets of baseline models (3B + 7B each) on the new dataset. The paper analyzed an existing dataset, ShotBench, refined some multiple choices to preserve mutual exclusivity, and randomly shuffled choices. The size of the dataset went from 3500 VQA to 961. The paper then benchmarked two sets of models on this refined dataset, and observed improved accuracy overall. The paper conducted further analysis on reasoning faithfulness, reasoning instruction adherence through qualitative and quantitative experiments, and proposed two scores: Faithful Reasoning Score (FRS) and Instruction Adherence Score (IAS) to reflect the reasoning process of the models. The paper conducted these evaluations using Qwen-3B as an automatic judge and observed that ShotVL lacks reasoning and instruction following capabilities while Qwen models demonstrated better performance.

**Strengths:**

1. The paper is clearly written and well structured
2. The paper contributed a refined dataset focusing on fine-grained cinematography terminology
3. The paper conducted thorough analysis and experimentations benchmarking two sets of VLMs on the new dataset, and revealed their limitations and strengths
4. The paper provided sufficient qualitative and quantitative analysis, easy-to-understand figures and examples to demonstrate the experiment results.

**Weaknesses:**

1. The paper conducted Qwen-as-judge to auto-evaluate the key reasoning capabilities of the baseline models. It would be great and important if these evaluates are validated through human evaluations.
2. The motivation of refining an existing dataset is focused around introducing and preserving mutual exclusivity among multiple choices. While it makes it 'cleaner' for humans to understand, it is possible that the refinement step made the task easier (as shown in Table 1). The question for the premise is: is mutual exclusivity actually necessary? Should the VLMs still be able to choose the right answer(s) even if they are not?
3. The paper could potentially be better suited for a more niche targeted group of audience
3. Questions below.

**Questions:**

1. Is there any reason that the dataset size drastically reduced from 3500 QAs to 961 QAs?
2. Table 3: if ShotVL models could achieve higher performance and in shorter time through 'Direct' predictions, compared to Qwen models, is it necessary to go through the reasoning steps, especially in real world applications?

Other minor questions:

3. Section 4.1 Reasoning Faithfulness -- Qualitative Analysis: the think and answer tags '?!'
4. Table 1 LC ShotVL-3b: why isn't 67.4 bold, compared to 65.7 for ShotVL-7B?
5. Ln 208: how did you define '16,7% confusion rate'

---

### Official Review · Reviewer_MnGU · 2025-11-01

**Soundness:** 3
**Presentation:** 3
**Contribution:** 2
**Rating:** 4
**Confidence:** 3

**Summary:**

This paper introduces RefineShot, a rigorously refined and extended benchmark for evaluating cinematography understanding in MLLMs. Motivated by the observation that the current standard benchmark, ShotBench, suffers from inconsistent option granularity and that its best-performing baseline, ShotVL, exhibits unreliable reasoning and poor instruction adherence, the authors first restructure ShotBench’s multiple-choice sets to enforce mutual exclusivity and dimensional consistency across eight cinematic attributes (e.g., lighting, framing, camera motion). An analysis of ShotVL is conducted, demonstrating that its high accuracy masks frequent mismatches between generated reasoning traces and final answers as well as dramatic performance drops under structured prompting. Finally, they augment the benchmark with an evaluation protocol that jointly measures task accuracy, and FRS/IAS indicator they proposed.

**Strengths:**

1. A systematic re-engineering of ShotBench that replaces ambiguous, cross-dimensional options with mutually exclusive choices of uniform granularity, yielding 961 revised questions and a demonstrably fairer evaluation set.

2. An analysis is conducted on the SOTA ShotVL family, exposing systematic unfaithfulness between its reasoning and answers and severe fragility under chain-of-thought or step-by-step prompts.

3. The introduction of complementary metrics—Faithful Reasoning Score (FRS) and Instruction Adherence Score (IAS)—that jointly assess task performance and core model competencies, enabling nuanced differentiation between superficial accuracy and genuine reasoning reliability.

**Weaknesses:**

1. The paper does not contribute any new data; the so-called ShotBench++ is merely a re-packaging of ShotBench, which considerably dilutes its novelty.

2. No new model architecture or training strategy is proposed. While abstaining from methodology is acceptable for a benchmark paper.

3. However, the absence of an in-house method then obliges the authors to evaluate an extensive suite of existing models (typically 15–20). Instead, only four models are tested.

4. Among these four, all are 3B or 7B parameter variants; 72 B checkpoints and proprietary APIs are ignored. If even these small-scale models already saturate ShotBench++, the task’s continued relevance is called into question.

5. Human is not in the loop. The paper does not report κ-scores or crowd-worker consistency for the 961 re-written options, leaving the community uncertain whether the “refined” labels are actually less ambiguous than the original ones.

**Questions:**

The present contribution is too slender—both in scientific advance and in empirical heft—to meet the bar of a top-tier venue such as ICLR. The authors **neither release new data nor propose a novel architecture, and the evaluation matrix is restricted to four small-scale models** (3B / 7B) while omitting larger checkpoints and commercial APIs. Consequently, ShotBench++ is better viewed as a light-weight extension of ShotBench than as a self-standing benchmark; indeed, **it is less comprehensive than the ShotBench, which at least supplied baseline models.**

Nevertheless, the paper does deliver two genuinely fresh insights: (i) the diagnosis that reasoning faithfulness and instruction adherence can collapse even when accuracy is high, and (ii) the technically simple yet powerful idea of “Benchmark Refinement’’—re-engineering option sets to guarantee mutual exclusivity and dimensional purity. This latter concept is ripe for generalisation: applying the same protocol to MMBench, MMMU, or MME could purge analogous ambiguities across the multimodal-evaluation ecosystem. If the authors were to broaden their scope to multiple flagship benchmarks and release a unified refinement toolkit with extensive human re-annotation, the resulting study would amass the critical mass of novelty, utility, and community impact expected of a top-tier publication.

---

### Note · Authors · 2025-11-14

I have read and agree with the venue's withdrawal policy on behalf of myself and my co-authors.